# Nordihydroguaiaretic Acid as a Novel Substrate and Inhibitor of Catechol *O-*Methyltransferase Modulates 4-Hydroxyestradiol-Induced Cyto- and Genotoxicity in MCF-7 Cells

**DOI:** 10.3390/molecules26072060

**Published:** 2021-04-03

**Authors:** Jin-Hee Kim, Jimin Lee, Hyesoo Jeong, Mi Seo Bang, Jin-Hyun Jeong, Minsun Chang

**Affiliations:** 1Yonsei Institute of Pharmaceutical Sciences, College of Pharmacy, Yonsei University, Incheon 21983, Korea; jheeeee@daum.net (J.-H.K.); organicjeong@yonsei.ac.kr (J.-H.J.); 2Graduate School of Biological Sciences, Sookmyung Women’s University, Seoul 04310, Korea; petia13@naver.com (J.L.); hyesoojeong@sookmyung.ac.kr (H.J.); 3Department of Biological Sciences, Sookmyung Women’s University, Seoul 04310, Korea; micro5513@naver.com; 4Research Institute for Asian Women, Sookmyung Women’s University, Seoul 04309, Korea; 5Research Institute for Women’s Health, Sookmyung Women’s University, Seoul 04310, Korea

**Keywords:** nordihydroguaiaretic acid, catechol-*O*-methyltransferase, catechol estrogens, DNA damage, breast cancer

## Abstract

Nordihydroguaiaretic acid (NDGA) is a major lignan metabolite found in *Larrea* spp., which are widely used in South America to treat various diseases. In breast tissue, estradiol is metabolized to the catechol estrogens such as 4-hydroxyestradiol (4-OHE_2_), which have been proposed to be cancer initiators potentially involved in mammary carcinogenesis. Catechol-*O*-methyltransferase (COMT) catalyzes the *O-*methylation of catechol estrogens to their less toxic methoxy derivatives, such as 4-*O*-methylestradiol (4-MeOE_2_). The present study investigated the novel biological activities of NDGA in relation to COMT and the effects of COMT inhibition by NDGA on 4-OHE_2_-induced cyto- and genotoxicity in MCF-7 human breast cancer cells. Two methoxylated metabolites of NDGA, 3-*O*-methylNDGA (3-MNDGA) and 4-*O-*methyl NDGA (4-MNDGA), were identified in the reaction mixture containing human recombinant COMT, NDGA, and cofactors. *K*_m_ values for the COMT-catalyzed metabolism of NDGA were 2.6 µM and 2.2 µM for 3-MNDGA and 4-MNDGA, respectively. The COMT-catalyzed methylation of 4-OHE_2_ was inhibited by NDGA at an IC_50_ of 22.4 µM in a mixed-type mode of inhibition by double reciprocal plot analysis. Molecular docking studies predicted that NDGA would adopt a stable conformation at the COMT active site, mainly owing to the hydrogen bond network. NDGA is likely both a substrate for and an inhibitor of COMT. Comet and apurinic/apyrimidinic site quantitation assays, cell death, and apoptosis in MCF-7 cells showed that NDGA decreased COMT-mediated formation of 4-MeOE_2_ and increased 4-OHE_2_-induced DNA damage and cytotoxicity. Thus, NDGA has the potential to reduce COMT activity in mammary tissues and prevent the inactivation of mutagenic estradiol metabolites, thereby increasing catechol estrogen-induced genotoxicities.

## 1. Introduction

Nordihydroguaiaretic acid (NDGA; 4-[4-(3,4-dihydroxyphenyl)-2,3-dimethylbutyl] benzene -1,2-diol) is a catecholic lignan mainly present in the five plant species that comprise *Larrea* [1]. For example, in *Larrea tridenta*, also known as “chaparral” and “creosote bush”, which is abundant in the deserts of Mexico and the southwest USA, NDGA accounts for approximately 10% of the dry weight of leaves [2]. The leaves of the creosote bush have been used in traditional medicine for the treatment of various diseases or symptoms, including arthritis, diabetes, inflammation, rheumatism, and pain [1]. We have previously shown that NDGA-containing extracts of *Larrea nitida* have phytoestrogenic properties and may therefore be applied as an alternative hormone replacement formulation in postmenopausal women [3]. In addition, NDGA has been shown to have a wide range of pharmacological activities, including the inhibition of lipoxygenase activity [4] and estradiol binding to sex steroid-binding protein [5] and the growth of viruses and tumors [6,7]. The anti-aging effects of NDGA are currently being tested using animal models [8]. Despite compelling preclinical proof on the health benefits of NDGA, or Larrea plant-based dieatary products containing NDGA, the major drawback for further clinical development is associated with toxicities, especially in the kidney and liver [9]. Thus, further evaluation is warranted to understand the types of toxicities and adverse outcome pathways associated with NDGA consumption.

Estradiol (E_2_) is enzymatically converted to the catechol estrogens, 2- and 4-hydroxyestradiol (2-OHE_2_ and 4-OHE_2_), via oxidative metabolism. Both these catechol estrogen metabolites can undergo further metabolism to generate reactive intermediates such as quinones, semiquinone radicals, and reactive oxygen species, which may react with biomacromolecules leading to cyto- and genotoxicity [10]. For example, 4-OHE_2_-quinones can form depurinating DNA adducts, a potential tumor initiating event in breast cancer and other human cancers, unless repaired [11]. Catechol *O-*methyl transferase (COMT) is a conjugative drug-metabolizing enzyme involved in the inactivation of many endogenous catechols by transferring a methyl group from *S*-adenosyl-L-methionine (SAM) to the hydroxyl group of the catechol substrates. It has been suggested that COMT plays a protective role in catechol estrogen-induced carcinogenesis, because both 2-OHE_2_ and 4-OHE_2_ are inactivated by COMT-catalyzed methylation [12]. Inhibition of COMT activity has been linked to carcinogenesis in animal models. Chronic treatment of Syrian hamsters with quercetin, a substrate and potent inhibitor of hamster renal COMT, significantly increased E_2_-induced renal tumorigenesis [13]. Genetic polymorphism of human COMT has been associated with increased cancer risk for estrogen-induced cancers owing to decreased COMT activity and inactivation of estrogen metabolites, although this is limited to specific ethnic groups or populations [14,15]. These data suggest that inhibition of COMT-mediated methylation of catechol estrogens by xenobiotics or genetic factors may facilitate the development of estrogen-induced tumors as a result of the accumulation of reactive catechol estrogen intermediates in target cells.

Several phytochemicals with catecholic structures have been studied for their inhibitory effects on COMT. Flavonols, isoflavones, catechins, alkaloids, and polyphenolic acids have been evaluated as COMT inhibitors and investigated for their implication in chemical-induced genotoxicity or their therapeutic value in treating Parkinson’s disease [16,17,18,19]. However, effects of phytochemicals belonging to the lignan class on COMT activity have not been reported yet. Considering NDGA-mediated toxicity, the potential market availability of Larrea-containing herbal preparations, and a role of COMT in carcinogenesis, it is critical to investigate the effects of NDGA on COMT activity and catechol estrogen-mediated cyto- and genotoxicity.

In the present study, we hypothesized that NDGA is a substrate for COMT and thus potentially inhibits the COMT-mediated metabolism of catechol estrogens. Inhibition of COMT would result in decreased detoxification of catechol estrogens, which in turn may lead to increased DNA damage and cytotoxicity. Therefore, we studied the implications of modulation of COMT activity by NDGA on catechol estrogen-induced DNA damage and cell death events using the alkaline comet assay, detection of AP depurination, and caspase-8/9 assays in the human mammary epithelial cancer cell line MCF-7.

## 2. Results

### 2.1. COMT-Mediated Metabolism of NDGA

As NDGA contains a catecholic structure, we investigated whether NDGA is a substrate of human soluble COMT. We previously identified two different *O*-methylated metabolites of NDGA present in the methanol extracts of *Larrea nitida* Cav leaves [3,20]. HPLC-DAD, LC-MS/MS, and NMR analyses showed that *O*-methylation occurred at either the C-3′ or C-4′ position of NDGA. These *O*-methyl NDGA metabolites were named 3-methoxy NDGA (3-MNDGA) and 4-methoxy NDGA (4-MNDGA), respectively (see Figure 1 for carbon numbering and chemical structures). In the present study, incubation of NDGA with human COMT and cofactors led to formation of two *O*-methylated metabolites of NDGA, 3-MNDGA and 4-MNDGA. Quantitation of 3-MNDGA was performed based on the standard curve using the reference compound that was isolated and identified from the extract of *Larrea nitida* Cav. [3]. A double reciprocal plot analysis of *O-*methylation at the C-3′ position revealed that *K*_m_ and *V*_max_ values were 2.57 μM and 15.7 µmol/(min·mg protein), respectively, and the rate of 3-*O-*methylation was 10.2 μmol/(min mg protein) (Table 1). Enzyme kinetic analysis of *O*-methylation of NDGA at the C-4′ position resulted in a *K*_m_ value of 2.19 μM. Vmax value was not calculated because the reference compound for 4-MNDGA was unavailable; thus, quantitation of 4-MNDGA amount based on peak area in HPLC was not performed. To verify the activities of the recombinant COMT prepared in this study in relation to 4-OHE_2_, enzyme kinetic analysis was also performed for the *O-*methylation of 4-OHE_2_. The *K*_m_ and *V*_max_ values for the formation of 4-MeOE_2_ were 37.9 µM and 3.26 µmol/min·mg protein, respectively. The affinity of NDGA to COMT and the maximal rate of methylation were 12- and 5-fold higher, respectively, than the respective values for the *O-*methylation of 4-OHE_2_. This implies that NDGA was a good substrate for COMT under our experimental conditions.

### 2.2. Inhibition of the COMT-Mediated O-Methylation of 4-OHE_2_ by NDGA

The inhibitory effects of NDGA on COMT activities were studied using 4-OHE_2_ as the COMT substrate. COMT-mediated *O-*methylation of 4-OHE_2_ was inhibited in an NDGA concentration-dependent manner and the IC_50_ for NDGA was calculated as 22.4 μM (Figure 2A). The *V*_max_ values of *O-*methylation of 4-OHE_2_ were markedly decreased as the concentration of NDGA increased (Figure 2B).

Such decreases in *V*_max_ in the presence of increasing NDGA concentrations indicated a substantial contribution by an uncompetitive mechanism of COMT inhibition. In addition, *K*_m_ increased as the concentration of NDGA increased, implying that inhibition type could be a competitive mechanism. Based on changes in both *K*_m_ and *V*_max_, we concluded that mixed inhibition of COMT occurs in the presence of NDGA. Using non-linear regression analysis, a *K*_i_ value was obtained as 15.7 μM. The amount of 4-MeOE_2_ was measured to study whether COMT activity in MCF-7 cells is affected by NDGA. Treatment of cells with either NDGA or Ro 41-0960, a known COMT inhibitor, resulted in significant decreases of 30%, 63%, and 88% in the formation of 4-MeOE_2_ with NDGA (25 µM), NDGA (50 µM), and Ro 41-0960 (10 μM), respectively (Figure 2C).

### 2.3. Molecular Docking Analysis of NDGA Bound to the Substrate Binding Pocket of COMT

NDGA, 4-OHE_2_, and dinitrocatechol (DNC) are positioned similarly towards the Mg^2+^ and SAM binding sites inside the substrate binding pocket of the human COMT active site (Figure 3A,B). Molecular docking was based on the crystal structure of a complex of COMT with DNC, a known COMT inhibitor (PDB code 3 BWM) [21]. Modeling data indicated that NDGA tends to adopt favorable binding modes within the COMT active sites, and this involves multiple hydrogen bonds and hydrophobic interaction (Figure 3C,D). Two pairs of dihydroxyl groups of each catechol present in NDGA engaged in five different hydrogen bonding interactions with Glu199, Asn170, Gly175, and Asp145 at the COMT active site (Figure 3E), whereas the dihydroxyl groups of both 4-OHE_2_ and DNC involved hydrogen bonds with Glu199 and Asn170 (Figure 3F,G). The hydrogens of the hydroxyl group on the phenyl group of NDGA interact with the oxygen of Glu199 through two hydrogen bonds at distances of 2.0 and 1.9 Å. The oxygen of the hydroxyl group on the phenyl group of NDGA interacts with the hydrogen of Asn170 at a distance of 2.3 Å. The hydroxyl groups on another phenyl group of NDGA interacted with the oxygen of Gly175 and the hydrogen of Asp145 at a distance of 2.4 and 2.7 Å, respectively (Figure 3E). The more flexible scaffold of NDGA may have allowed the formation of more molecular contacts by NDGA with the amino acids Asp145 and Gly175, which are not involved in the interaction with 4-OHE_2_. The presence of Trp143 at the COMT active site seemed to facilitate the formation of hydrophobic contacts with SAM, NDGA, 4-OHE_2_, and/or DNC (Figure 3C,D). Docking scores were calculated as 7.4158, 4.5549, and 5.9104 for NDGA, 4-OHE_2_, and DNC, respectively, indicating that NDGA theoretically possessed a higher binding affinity than that of 4-OHE_2_ or DNC toward the COMT active site. The higher docking score for NDGA was consistent with the low *K*_m_ and high *V*_max_ values of NDGA in comparison to those of 4-OHE_2_. Modeling data suggest that NDGA might be positioned in a more stabilized conformation than 4-OHE_2_ or DNC. Thus, NDGA might serve as either a good substrate or an inhibitor for COMT.

### 2.4. Effects of NDGA on 4-OHE_2_-Induced Cytotoxicity and Apoptosis in MCF-7 Cells

Next, we determined the effects of NDGA on 4-OHE_2_-mediated cell death via modulation of COMT activity. Significant cell death was not observed in the cells treated with either 25 or 50 μM NDGA compared to that in the DMSO-treated samples. Treatment of MCF-7 cells with 4-OHE_2_ alone caused a 30% decrease in cell viability, and that with 4-OHE_2_ and either 25 or 50 μM NDGA resulted in a 45% reduction in cell viability (Figure 4A). These data imply that co-treatment of NDGA is associated with increased 4-OHE_2_-induced cytotoxicity, showing no significant differences in cell viability between the two concentrations of NDGA. The apoptosis level was clearly increased in cells treated with both NDGA and 4-OHE_2_ by 1.5- to 1.7-fold compared to that in cells treated with only 4-OHE_2_ as determined based on caspase-8 levels (Figure 4B). The levels of caspase-9 were also increased by approximately 1.7-fold upon co-treatment of NDGA and 4-OHE_2_ (Figure 4B). There were no significant differences in the increase between caspase-8 and caspase-9 levels or between concentrations (25 μM and 50 μM) of NDGA.

### 2.5. Effects of NDGA on 4-OHE_2_-Induced DNA Damage

To investigate the implications of COMT inhibition by NDGA, effects of NDGA on 4-OHE_2_-induced DNA damage were studied in MCF-7 cells. Both the genomic AP site detection assay and comet assay showed that incubation of MCF-7 cells with NDGA alone did not cause DNA damage when compared with the DMSO-treated cells. Cells treated with 4-OHE_2_ alone showed a 1.5- and 2.1-fold increase in DNA damage in the AP site detection (Figure 5A) and comet (Figure 5B) assays, respectively. Compared to 4-OHE_2_ treatment alone, concomitant incubation with NDGA showed an increase of 1.6- (NDGA 25 µM) and 1.8-fold (NDGA 50 µM) in 4-OHE_2_-induced DNA damage measured by AP site detection assays (Figure 5A). Co-treatment of NDGA with 4-OHE_2_, compared to that of 4-OHE_2_ alone, also increased the %DNA tail in the comet assay by 1.2 (NDGA 25 µM) and 1.4-fold (NDGA 50 µM).

## 3. Discussion

Lignans are a large group of polyphenols found in plants, particularly in seeds, whole grains, and vegetables [22]. They are derived from oxidative dimerization of more than two phenylpropanoid units. Numerous studies have shown the health benefits of lignans, such as anti-inflammatory, antibiotic, anticancer, and antioxidant effects. NDGA belongs to the class of lignans with catecholic structures (Figure 1). It has a wide range of biological activities [23]. To date, no studies have reported the biological effects of NDGA on COMT. The present study showed that NDGA is a COMT substrate with *K*_m_ values of 2.19 and 2.57 µM for the generation of 3-*O*-methyl and 4-*O*-methyl derivatives of NDGAta, respectively (Table 1). In addition, utilizing Lineweaver–Burk analysis, our study showed that NDGA is a mixed-type inhibitor of COMT. Using cytosolic fractions derived from human tissues of the breast, liver, or placenta; cultured human cancer cells; or hamster kidney tissues as enzyme sources, several studies have shown that flavonols (e.g., quercetin), isoflavones (e.g., genistein), catechins (e.g., (-)-epigallocatechin-3-gallate), and phenolic acids (e.g., chlorogenic acid) are also COMT inhibitors with a mixed-type inhibition mode with IC_50_ values at micromolar concentration ranges [16,18,24,25]. In addition, these phytochemicals were identified as COMT substrates owing to their catecholic structures. Consistent with these previous studies on catecholic phytochemicals and their effects on COMT, our study showed that NDGA is both a good substrate and inhibitor of COMT. This study is the first to demonstrate the novel activity of a lignan-type phytochemical toward COMT.

The computer modeling study supported the notion that NDGA fits inside the COMT active site, allowing NDGA to act as a good substrate and inhibitor for COMT (Figure 3). Results showed that NDGA adopted a stable conformation through hydrogen bond networking with Asp145, Asn170, Gly175, and Glu199 in the active site of COMT. Regarding the COMT-inhibitory potency and efficacy of phytochemicals presented in the previous studies, most catecholic phytochemicals show IC_50_ values ranging between hundreds in the nanomolar to the micromolar concentration; in our study, the IC_50_ value of NDGA for COMT-mediated *O*-methylation of 4-OHE_2_ was found to be 22.4 µM. It seems that molecular interactions between phytochemicals and COMT at the active site play a significant role in the inhibitory potency. The lack of hydrogen bond interactions observed in the case of the non-catecholic indole alkaloid *Z*-vallesiachotamine explains its high IC_50_ value of 196 µM [26]. Submicromolar range IC_50_ values observed with chlorogenic acid and (-)-epigallocatechin-3-gallate were explained by the more stable conformation through the hydrogen bonding networks between structurally flexible hydroxyl groups of the ligand and Asn170 or Lys144 at the COMT active site [17,25]. In the case of NDGA, five hydrogen bonding interactions were identified in the molecular docking studies (Figure 3B). These interactions were possible owing to the unique chemical structure of NDGA, which is characterized by a free rotation of the carbon backbone and four terminal hydroxyl groups. The favorable hydrogen bonding interactions are believed to contribute to the affinity toward COMT and the inhibitory potency presented by NDGA in our study.

We further investigated whether the inhibitory potential of NDGA observed in our study using a recombinant COMT was also true for the whole cell system and whether COMT inhibition by NDGA affects catechol estrogen-mediated toxicity. It was demonstrated that the decreased biotransformation of catechol estrogens because of COMT inhibition by Ro 41-0960 is associated with increased oxidative DNA damage in MCF-7 cells [27]. van Duursen et al. demonstrated that 4-OHE_2_-mediated DNA damage was exacerbated in MCF-7 cells by co-treatment with COMT inhibitors such as Ro 41-0960 and/or quercetin, presumably by decreasing the detoxification potential of COMT on catechol estrogens [16]. Our results showed that, first, the COMT-catalyzed *O*-methylation of 4-OHE_2_ was significantly decreased in the presence of either NDGA or Ro 41-0960 in MCF-7 cells (Figure 2C); and second, NDGA increased 4-OHE_2_-induced DNA damage, cell death, and apoptosis in the same cell line. These data imply that NDGA-mediated COMT inhibition is associated with a decrease in inactivation of the potentially toxic 4-OHE_2_ and a significant increase in 4-OHE_2_-induced cyto- and genotoxicity.

Various health benefits of NDGA have been reported. For example, NDGA can inhibit growth factor signaling to help breast cancer treatment or prevention [28]. In addition, it has weak ER agonistic activity, which would indicate that NDGA could be useful as a phytoestrogen-based functional food ingredient. However, clinical development has been delayed due to its toxicities on the major organs. We previously showed that NDGA and MNDGA have the potential to inhibit the activities of hepatic drug-metabolizing enzymes and form reactive intermediates, which would lead to unexpected adverse effects [20]. The present study adds new evidence toward the potential toxicity of NDGA through the inhibition of COMT activity, which is associated with increased toxicities of catechol estrogens. Although there is no clear information regarding the clinically relevant concentration of NDGA to exert either pharmacological or toxicological effects in vivo, some biological effects of NDGA in cancer cell models were studied at concentrations of 20–100 µM, which are within our experimental conditions [23]. Further in vivo, in vitro, and molecular investigations of the roles of NDGA in catechol estrogen-associated carcinogenesis would provide more solid evidence. Regardless, our study provides an insight into the adverse outcome pathway associated with the consumption of NDGA.

## 4. Materials and Methods

### 4.1. Chemicals and Reagents

All chemicals, solvents, and reagents were purchased from Sigma (St. Louis, MO, USA) or Burdick & Jackson (Morristown, NJ, USA) unless stated otherwise. NDGA was purchased from Santa Cruz (Dallas, TX, USA). 3-MNDGA was provided by the Korea Research Institute of Bioscience and Biotechnology (Ochang, Korea). NDGA and 3-MNDGA were dissolved in dimethyl sulfoxide (DMSO). Recombinant human soluble COMT protein was prepared as the hexahistidine-tagged protein as described previously [1]. This enzyme was stored in 10 mM Tris (pH 7.8) containing 1 mM EDTA and 0.2 mM DTT at −80 °C.

### 4.2. Enzyme Assays and Kinetic Measurements

COMT activity was determined by analyzing the enzyme incubation samples using high-performance liquid chromatography diode array detection (HPLC-DAD). The HPLC system (Agilent, Santa Clara, CA, USA) consisted of an Agilent 1200 binary pump, DAD, and Agilent 1260 autosampler. A Kinetex^®^ C_18_ reverse-phase column (5 µm, 4.6 × 150 mm) (Phenomenex, Torrance, CA, USA) protected by a KrudKatcher Ultra HPLC in-line filter (Phenomenex) was used for the separation. Mobile phase A was water with 0.1% (*v*/*v*) formic acid, whereas mobile phase B was acetonitrile. The solvent flow rate was 1 mL/min, and the column temperature was set at 35 °C. The DAD was set at a wavelength of 280 nm. The typical reaction mixtures (total volume 200 µL) contained COMT (2 µg of the recombinant enzyme or 100 µg of cell homogenate), SAM (0.2 mM), MgCl_2_ (5 mM), and NDGA at concentrations ranging from 5 **μ**M to 50 **μ**M in sodium phosphate buffer (50 mM, pH 7.8). The inhibitory activity of NDGA on the COMT-mediated *O-*methylation of 4-OHE_2_ was examined using a mixture of COMT, 4-OHE_2_ (50 µM), and various concentrations of NDGA in the presence of cofactors. The reaction was initiated by the addition of SAM followed by incubation in a shaking incubator at 37 °C for 60 min and stopped by the addition of 20 µL of trichloroacetic acid. After centrifugation (12,000× *g* for 10 min at 4 °C), the supernatants (20 µL) were subjected to HPLC-DAD analysis for quantitation of 4-*O*-methylestradiol (4-MeOE_2_).

### 4.3. Molecular Docking Study

Structures of NDGA, 4-OHE_2_, or DNC as ligands bound to the human COMT active site were constructed using the SYBYL-X2.1.1 molecular modeling software (Tripos Inc., St. Louis, MO, USA) and energy was minimized by the Powell method using the Gasteiger−Marsili charge and the Tripos force field [29]. The crystal structure of DNC-bound COMT was obtained from the Protein Data Bank (PDB code 3BWM) [21]. All water molecules from the crystal structures were removed, whereas the missing hydrogen atoms were added to the structures. Docking was performed using Surflex-Dock in SYBYL-X2.1.1 (Tripos Inc.) [30]. For the protein, the protocol that characterizes the binding site of the enzyme was generated using a ligand/substrate-based approach. All other parameters were set to default. Either NDGA or 4-OHE_2_ in 3BWM was subjected to a redocking process and the best docked conformation of the NDGA-bound substrate binding pocket was superimposed on that of the 4-OHE_2_-bound substrate binding pocket. Finally, the Surplex-Dock scoring function was the sum of the hydrophobic, polar, repulsive, and entropic terms including crash and solvation over the appropriate atom pairs.

### 4.4. Cell Culture

The MCF-7 human breast cancer cell line was purchased from the American Type Culture Collection (Manassas, VA, USA). The cells were maintained in Dulbecco’s modified Eagle’s essential medium containing 10% fetal bovine serum and 1% antibiotics/antimycotics at 37 °C in a 5% CO_2_ atmosphere. To determine COMT activity, the cell homogenates were prepared by washing cells with PBS, then trypsinized and homogenized in 10 mM sodium phosphate buffer (pH 7.4) containing 0.5 mM dithiothreitol. The homogenates were centrifuged at 1000× *g* for 10 min at 4 °C. The supernatant was used to determine COMT activity.

### 4.5. Cytotoxicity and Apoptosis Assays

MCF-7 cells were treated with 4-OHE_2_, NDGA, or both for 48 h. Cell viability was determined using the 3-(4,5-dimethylthiazol-2-yl)-2,5-diphenyltetrazolium bromide (MTT) assay as described previously [31]. Apoptosis was determined by detecting increased levels of caspase-8 and -9. For this, cells treated with appropriate compounds for 48 h were washed, lysed with 200 µL of chilled cell lysis buffer (Promega, Madison, WI, USA), and incubated on ice for 10 min. After centrifugation at 10,000× *g* for 1 min at 4 °C, the supernatant was subjected to caspase detection assays using human caspase-8 and -9 ELISA kits (Abcam, Cambridge, UK). Levels of caspase were quantitated by reading the final product at 405 nm using a SpectraMax i3x plate reader (Molecular Devices, San Jose, CA, USA).

### 4.6. Alkaline Single-Cell Gel Electrophoresis (Comet) Assay

The effect of COMT inhibition on DNA damage caused by 4-OHE_2_ was determined using the alkaline single-cell gel electrophoresis (comet) assay as described previously by Singh et al., with some modifications [32]. For this assay, 1 × 10^6^ MCF-7 cells were cultured in 100 mm cell culture dishes and exposed to appropriate test compounds and incubated in a 5% CO_2_ atmosphere at 37 °C. After incubation for 48 h, the medium was removed and the cells were washed three times with PBS, trypsinized, harvested, and counted. The cells were spun down at 1000× *g* for 3 min. Cells were gently resuspended in PBS (200 µL) and a 10 µL aliquot was added to 90 µL of warm low-melting agarose. This mixture (50 µL) was immediately pipetted onto CometSlide^TM^ (Trevigen, Gaithersburg, MD, USA) slides. These slides were placed flat in a refrigerator at 4 °C for 10 min, and then immersed overnight in pre-chilled lysis solution (2.5 M NaCl, 0.1 M EDTA, 10 mM Tris, 34 mM *N*-lauroylsarcosinate sodium salt, 1% Triton X-100, pH 10.0) at 4 °C. The slides were immersed in freshly prepared alkaline solution (0.2 M NaOH and 1 mM EDTA, pH > 13) for 20 min and then electrophoresed in alkaline solution for 30 min at 20 V. The slides were then washed three times with a neutralization buffer (0.4 M Tris, pH 7.5) and dehydrated in 70% ethanol for 5 min at RT. The slides were stained with SYBR Green for 12 h at RT. Image analysis was performed under an Olympus IX71 fluorescence microscope (Tokyo, Japan) using a 20× objective and a filter of 450–490 nm. Approximately 150 cells per slide were selected for the OpenComet analysis [33].

### 4.7. Apurinic/apyrimidinic Site Assay

To study the effect of COMT inhibition by NDGA on the oxidative DNA damage caused by 4-OHE_2_, AP sites were quantitated using an OxiSelect^TM^ oxidative DNA damage quantification kit (Cell Biolabs, San Diego, CA, USA) according to the manufacturer’s instructions, using the aldehyde-reactive probe DNA standards to quantify the number of genomic AP sites.

### 4.8. Statistical Analysis

Experiments using cells and enzyme incubations were performed in triplicate and repeated at least three times. Results are representative of at least three independent experiments. All results are presented as mean ± standard error (SE) values. An F-test was used to determine if variances among groups were equal. If sample variances were homogenous, analysis of variance (ANOVA) was performed followed by Bonferroni posthoc tests using Prism version 9.0 (GraphPad Software, San Diego, CA, USA). Statistical comparisons between the control and treated groups were performed using Student’s *t*-test. Differences in results among data sets were considered statistically significant when the *p*-value was less than 0.01. Significant differences are indicated by appropriate symbols in the figures. Calculation of IC_50_ and *K*_i_ values was performed using Prism version 9.0.

## 5. Conclusions

NDGA was identified as a substrate and inhibitor for COMT. NDGA-mediated COMT inhibition is associated with catechol estrogen-induced cyto- and genotoxicities in MCF-7 cells. Our data provide molecular insights into potential toxic mechanisms by which NDGA aggravates catechol estrogen-induced toxicities.

## Figures and Tables

**Figure 1 molecules-26-02060-f001:**
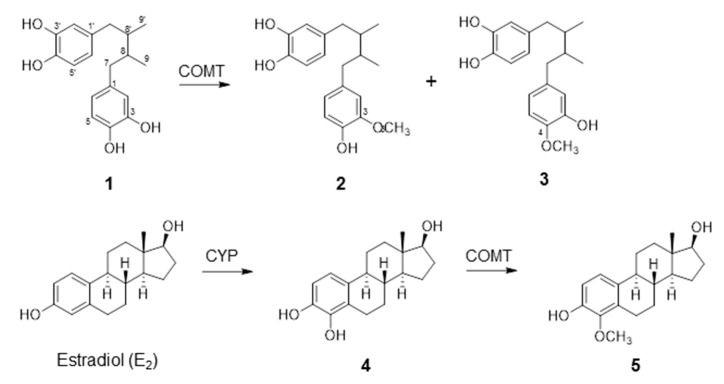
Schemes for the catechol *O*-methyl transferase (COMT)-mediated metabolism of nordihydroguaiaretic acid (NDGA) and 4-hydroxyestradiol (4-OHE_2_). NDGA (**1**) is a substrate for the COMT to generate two different *O*-methylated products at either 3-hydroxyl group (**2**) or 4-hydroxyl group (**3**). 4-OHE_2_ (**4**), a catechol estrogen metabolite of E_2_, is biotransformed to 4-*O*-methylated hydroxyestradiol (4-MeOE_2_) (**5**).

**Figure 2 molecules-26-02060-f002:**
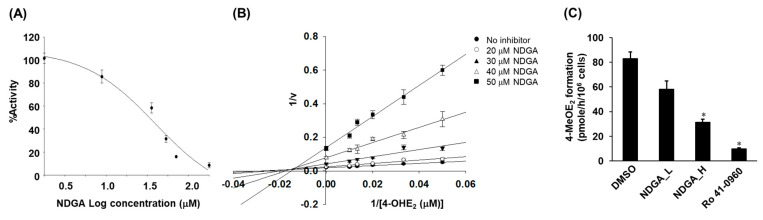
Inhibition of catechol *O*-methyl transferase by nordihydroguaiaretic acid. COMT activity was calculated as a percentage with the peak area for 4-*O*-methylated hydroxyestradiol obtained in the HPLC-DAD analysis set as 100 with no inhibitor present. The IC_50_ value of NDGA-mediated COMT inhibition of 4-hydroxyestradiol methylation was obtained by incubating COMT in the presence of various concentrations of NDGA and 4-OHE_2_ (50 µM) (**A**). The inhibition kinetic parameters were further analyzed using a double-reciprocal plot (**B**). Inhibition of COMT-catalyzed 4-OHE_2_ methylation in whole cell (MCF-7) lysate treated with either NDGA or Ro 41-0960 (10 μM) (**C**). NDGA_L indicates 25 µM of NDGA, whereas NDGA_H indicates 50 µM NDGA-treated cell samples. Each data point is representative of three independent experiments. Data are shown as the mean values of three determinations ± SE. Each experiment was independently repeated at least three times. An asterisk indicates *p* < 0.001 between compound-treated and DMSO-treated cells.

**Figure 3 molecules-26-02060-f003:**
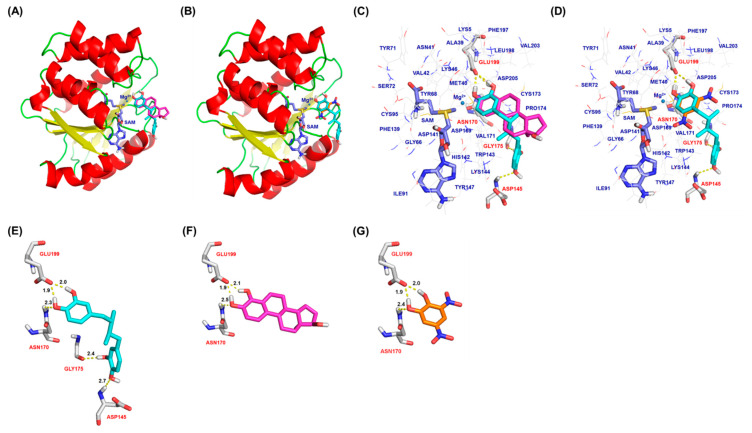
Computational models for human catechol *O*-methyl transferase bound with 4-hydroxyestradiol, dinitrocatechol (DNC), and nordihydroguaiaretic acid. Binding conformation for COMT bound with NDGA and 4-OHE_2_ (**A**), and NDGA and DNC (**B**) in a ribbon representation. The secondary structure backbone of human COMT (PDB code 3BWM) is represented as a cartoon and colored red for the helix, yellow for the sheet, and green for the loop. *S*-adenosyl-L-methionine (SAM; violet), Mg^2+^ (blue sphere), NDGA (cyan), 4-OHE_2_ (magenta), and DNC (orange) all bound on the protein surface of the binding site of human COMT. Superimpositions of NDGA (cyan) and 4-OHE_2_ (magenta), and NDGA and DNC (orange), at the human COMT active site are shown as stick representations (**C**) and (**D**), respectively. The amino acid residues involved in the key hydrogen bonding interactions are depicted by atom type: carbon atoms in gray, nitrogen atoms in blue, sulfur atoms in yellow, and oxygen atoms in red. Polar hydrogen bonds are shown as yellow dashed lines. A comparative view of the hydrogen bond interactions between COMT and NDGA (cyan) (**E**), 4-OHE_2_ (magenta) (**F**), and DNC (orange) (**G**) is shown as a stick representation. Numbers presented adjacent to yellow dashed lines represent the calculated length of the hydrogen bond in Å.

**Figure 4 molecules-26-02060-f004:**
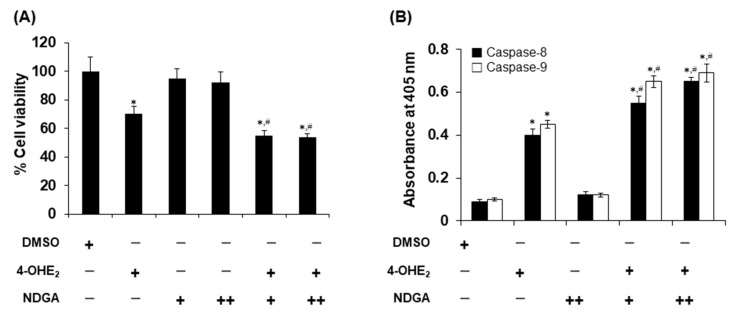
Effects of nordihydroguaiaretic acid on 4-hydroxyestradiol-induced cell death and apoptosis. MCF-7 cells were treated with 4-OHE_2_ (50 µM) and NDGA (either 25 or 50 µM). Cell viability was measured using the MTT assay (**A**) and apoptosis was determined by the detection of increased caspase-8 and -9 levels using human caspase ELISA kits (**B**). The label “+” represents results obtained from samples treated with 25 µM of NDGA, whereas “++” means samples treated with 50 µM NDGA. Data are shown as mean values of three measurements ± SE. Each experiment was independently repeated at least three times. * indicates *p* < 0.001 between compound-treated and DMSO-treated cells. # indicates significantly differences between samples treated with 4-OHE_2_ and 4-OHE_2_ + NDGA (*p* < 0.05).

**Figure 5 molecules-26-02060-f005:**
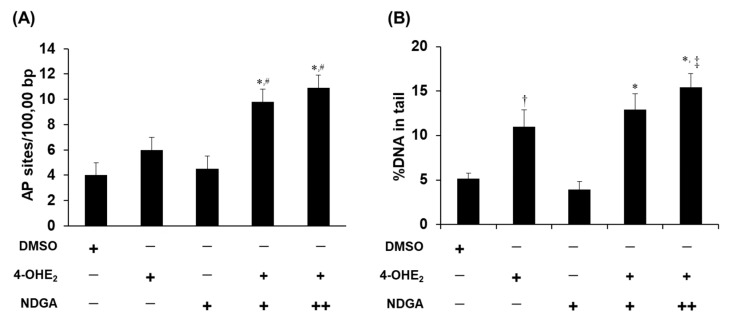
Catechol estrogen-induced DNA damage in MCF-7 cells and the effect of catechol *O*-methyl transferase inhibition. MCF-7 cells were incubated with 4-OHE_2_ (25 mM) and/or NDGA (either 25 or 50 μM). AP sites per 10,000 bp were measured using OxiSelect^TM^ oxidative DNA damage quantification kit (**A**). The comet assay was performed twice in duplicate, and approximately 150 comets per slide were analyzed. Data are represented as mean %DNA in tail (comet tail length × % of total DNA in the comet tail) of four slides in the DNA isolated from MCF-7 cells treated with either 4-OHE_2_ alone or both 4-OHE_2_ and NDGA (**B**). The label “+” represents results obtained from samples treated with 25 µM of NDGA, whereas “++” means samples treated with 50 µM NDGA. Data are shown as mean values of three measurements ± SE. Each experiment was independently repeated at least three times. * indicates significant differences between DMSO and compound-treated cells (*p* < 0.001). † indicates significantly differences between DMSO and compound-treated samples (*p* < 0.05). # indicates significantly differences between samples treated with 4-OHE_2_ and 4-OHE_2_ + NDGA (*p* < 0.05). ^‡^ indicates significantly differences between samples treated with 4-OHE_2_ and 4-OHE_2_ + NDGA (*p* < 0.01).

**Table 1 molecules-26-02060-t001:** Kinetic parameters in *O*-methylation of nordihydroguaiaretic acid (NDGA) and 4-hydroxyestradiol (4-OHE_2_) by human recombinant soluble catechol *O*-methyl transferase (COMT).

Substrate	Product	Rate of *O-*Methylation (µmol/mg Protein/min)	*K*_m_ (µM)	*V*_max_ (µmol/mg Protein/min)	*V*_max_/*K*_m_
NDGA ^a^	3-*O*-methyl NDGA (3-MNDGA)	10.2	2.57	15.7	6.12
NDGA ^b^	4-*O*-methyl NDGA (4-MNDGA)	N.D ^c^	2.19	N.D.	N.D
4-OHE_2_	4-*O*-methylestradiol (4-MeOE_2_)	2.62	37.9	3.26	0.086

^a^ Formation of 3-*O*-methyl NDGA (3-MNDGA); ^b^ formation of 4-*O*-methyl NDGA (4-MNDGA); ^c^ not determined because the reference for 4-*O*-methyl NDGA was not available.

## Data Availability

The data presented in this study are available in this article.

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
