# Peer review of "Nordihydroguaiaretic Acid as a Novel Substrate and Inhibitor of Catechol O-Methyltransferase Modulates 4-Hydroxyestradiol-Induced Cyto- and Genotoxicity in MCF-7 Cells"

_molecules, 2021, doi:10.3390/molecules26072060_

Round 1
Reviewer 1 Report
The paper by Jin Hee Kim et al. reports that the lignan metabolite NDGA could act as both substrate and inhibitor of COMT, therefore potentially disturbing the detoxification of 4-hydroxyestradiol.
This study might worth publication in Molecules, but the text has to be improved to afford better understanding and eliminate what looks like mistakes.
Overall, the english language needs to be improved in some parts of the manuscript.
Among raised problems:
Abstract, line 30: “molecular docking studies” predicted but not “showed”.
Introduction, lines 50 to 52: this sentence needs to be rewritten
Lines 73-74: “which confers the decrease in COMT” is not clear.
Lines 98 to 101: according to “As described previously, there are two…metabolites of NDGA that are derived from O-methylation…”, NDGA has already been shown to be a substrate of COMT, but this fact is supposed to be first demonstrated in the present paper… In addition, the reference given at the end of this sentence (i.e. [26]) does not concern NDGA.
Lines 105 to 109: this sentence is not at all understandable: “Enzyme kinetics… was (!) performed (?) based on the area of the peak (very vague)…” and all the rest of the sentence…
Lines 133 to 137: these two sentences have to be made clearer.
Lines 137 to 139: the same sentence repeated twice.
Line 140: what is the concentration of Ro41-0960 used in these experiments ?
Figure 2B: no error bars.
Lines 161 to 164: this sentence is very difficult to follow. Reword “The hydrogens bound to the oxygens on the hydroxyl groups of a phenyl group of NDGA…”. Also, not possible to say that “The hydrogens…interacted…with the hydrogen of Asn270…”.
Line 168: “Dock scores” should be “Docking scores” ?
Lines 196-197: how the Authors explain the absence of difference in effect between 25 and 50 microM of NDGA ?
Figure 5A and 5B: the “++” and “+” for NDGA have been reversed.
Line 283: “Figure 4C” should be Figure 2C ?
Lines 321 to 324: the description is not clear: “…various concentrations of NDGA to determine the level of COMT activity required to convert NDGA…”: could the Authors explain this point ?
Line 336: “The crystal structure of inhibitor-bound COMT…”: this inhibitor has to be indicated. It is dinitrocatechol. It should be interesting to compare the docking position of NDGA with the experimental position of dinitrocatechol (with presentation in Figure 3).
Line 353: “dithiothreitol” ?
Author Response
Dear Reviewer 1:
We greatly appreciate your detailed review of our manuscript and your valuable comments, which have helped us in improving our manuscript. We have carefully considered the comments and accordingly revised the main manuscript. Our changes to specific paragraphs, text, words, or numbers in the revised manuscript are shown in red, and the revised manuscript has been uploaded separately. Our point-by-point responses to the comments (in blue bold fonts) in a tabular form are found in a separate as an attachment.

Reviewer 2 Report
The paper "Nordihydroguaiaretic Acid as a Novel Substrate and Inhibitor 2 of Catechol O-Methyltransferase Modulates 4-Hydroxyestra- 3 diol-Induced Cyto- and Genotoxicity in MCF-7 cells" by Jin Hee Kim et al. is focussed on the investigation of the novel biological activities of Nordihydroguaiaretic acid (NDGA) toward Catechol-O-methyltransferase (COMT). In addition, the effects of COMT inhibition by NDGA on 4-hydroxyestradiol-induced cyto- and genotoxicity in MCF-7 human breast cancer cells have been evaluated.
The paper addresses a potentially significant research topic, namely the lignan metabolites biological effects.
The investigation is carried out with proper methodologies, and the conclusions are supported by the experimental evidences.
minor points:
Even if the mixed type inhibition is evident, the authors should use other methods to check it. At least they should include error bars in the double reciprocal plot, since it is known this latter is affected by quite high error propagation both on x and y values (see references).
references:
1. "The Problem with Double Reciprocal Plots", Raymond S. Ochs, Current Enzyme Inhibition, 2010, 6, 164-169
2. "An Introduction to Error Analysis: The Study of Uncertainties in Physical Measurements", John R. Taylor, Univ Science Books,1997, ISBN-10 : 9780935702750
Author Response
Dear Reviewer 2:
Thank you for your detailed review of our manuscript and the valuable comments, which were helpful in improving our manuscript. We have revised the main manuscript in accordance with the Reviewer’s comments, and the modified texts, numbers or paragraphs have been indicated in blue. We have uploaded the revised manuscript separately and provided our responses to reviewer’s comments in this letter.
Reviewer's point 1:
Even if the mixed type inhibition is evident, the authors should use other methods to check it. At least they should include error bars in the double reciprocal plot, since it is known this latter is affected by quite high error propagation both on x and y values (see references).
Response 1:
We generated graphs with appropriate error bars. The updated figure 2B (page 4) is presented in the revised manuscript. Our changes according to your comments are shown in blue in the revised manuscript.
- We appreciate the reviewer’s comment for providing an excellent reference for improving the kinetic analysis. Undoubtedly, analysis of the inhibition type using a double reciprocal plot can provide misleading information owing to the mathematical and intuitive errors inherent in this analytic method, as described by Ochs et al. (cited in the references). Nonetheless, this method is still widely used to extract kinetic parameters and provide visual assessment of the type of inhibition. We would like to retain the double reciprocal plot (a newly generated plot figure with error bars; Figure 2B).
- As the reviewer suggested, we have analyzed our data using non-linear regression analysis, a widely used method. Our analysis was performed using the “non-linear regression analysis/enzyme kinetics inhibition” function of Prism version 9.0 (GraphPad Software, San Diego, CA). Analysis resulted in an acceptable fit for the mixed model inhibition with a Ki value of 15.7 µM at 95% CI. We have added the result of non-regression analysis in the revised manuscript. Accordingly, the method to obtain Ki values was described as follows:
[Line 138-139] “Using non-linear regression analysis, a Ki value was obtained as 15.7 µM.”
[Line 374-375, 4.8 Statistical Analysis]
In the original manuscript: Calculation of IC50 value was performed using Prism version 4.0.
In the revised manuscript: Calculation of IC50 and Ki values was performed using Prism version 9.0.

Round 2
Reviewer 1 Report
It should be indicated in the 2.3 paragraph that docking experiments have been done based on 3BWM, writing that it is the crystal structure of a complex of COMT with dinitrocatechol (not sufficient in just the Materials and Methods section (to note, dinitrocatechol is not well written line 332)). It is also not clear why the Figure R1 presented in the response to Reviewers was not included or combined to Figure 3 in the new manuscript version.
Author Response
Dear Reviewer:
Our changes to specific paragraphs, text, words, or numbers in the revised manuscript are shown in purple, and the revised manuscript has been uploaded separately. Our point-by-point to the comments are described in this letter.
Point 1. It should be indicated in the 2.3 paragraph that docking experiments have been done based on 3BWM, writing that it is the crystal structure of a complex of COMT with dinitrocatechol (not sufficient in just the Materials and Methods section
Response 1
We agree with the Reviewer’s comment in that description on the source of docking study should be provided in the Result section 2.3.
We fixed the sentence in Line 149-150 in the revised manuscript as follows:
“Molecular docking was based on the crystal structure of a complex of COMT with DNC, a known COMT inhibitor (PDB code 3 BWM) [21]”
Point 2. (to note, dinitrocatechol is not well written line 332)).
Response 2
Thank you for pointing out typo. We changed to “DNC” in line 308 in the revised manuscript, since chemical full name “dinitrocatechol” appeared ahead (Section 2.3) and abbreviation “DNC” was already introduced.
Point 3. It is also not clear why the Figure R1 presented in the response to Reviewers was not included or combined to Figure 3 in the new manuscript version.
Response 3
The docking study result to compare the DNC with NDGA, which was provided only in Round 1 Response letter, was combined to Figure 3 as a reviewer suggested. Results of DNC-bound COMT appeared in Figure 3B, 3D, and 3G. We believe that addition of these results would support that NDGA exerts a good affinity toward COMT active site.
Accordingly, Result 2.3 section as well as Figure 3 legend have been modified to contain information regarding the modeling data performed with DNC as follow:
[Line 146 ~ 173]
2.3 Molecular Docking Analysis of NDGA Bound to the Substrate Binding Pocket of COMT
NDGA, 4-OHE2, and dinitrocatechol (DNC) are positioned similarly towards the Mg2+ and SAM binding sites inside the substrate binding pocket of the human COMT active site (Figure 3A and 3B). Molecular docking was based on the crystal structure of a complex of COMT with DNC, a known COMT inhibitor (PDB code 3 BWM) [21]. Modeling data indicated that NDGA tends to adopt favorable binding modes within the COMT active sites, and this involves multiple hydrogen bonds and hydrophobic interaction (Figure 3C and 3D). Two pairs of dihydroxyl groups of each catechol present in NDGA engaged in five different hydrogen bonding interactions with Glu199, Asn170, Gly175, and Asp145 at the COMT active site (Figure 3E), whereas the dihydroxyl groups of both 4-OHE2 and DNC involved hydrogen bonds with Glu199 and Asn170 (Figure 3F and 3G). ….. contacts with SAM, NDGA, 4-OHE2 and/or DNC (Figure 3C and 3D). Docking scores were calculated as 7.4158, 4.5549, and 5.9104 for NDGA, 4-OHE2, and DNC, respectively, indicating that NDGA theoretically possessed a higher binding affinity than that of 4-OHE2 or DNC toward the COMT active site. The higher docking score for NDGA is consistent with the low Km and high Vmax values of NDGA in comparison to those of 4-OHE2. Modeling data suggest that NDGA might be positioned in a more stabilized conformation than 4-OHE2 or DNC. Thus, NDGA might serve as a either good substrate or an inhibitor for COMT.
[Page 5] Figure 3 legend
Figure 1. Computational models for human catechol O-methyl transferase (COMT) bound with 4-hydroxyestradiol (4-OHE2), dinitrocatechol (DNC), and nordihydroguaiaretic acid (NDGA). Binding conformation for COMT bound with NDGA and 4-OHE2 (A), and NDGA and DNC (B) in ribbon representation. The secondary structure backbone of human COMT (PDB code 3BWM) is represented as a cartoon and colored in red for the helix, yellow for the sheet, and green for the loop. S-adenosyl-L-methionine (SAM; violet), Mg2+ (blue sphere), NDGA (cyan), 4-OHE2 (magenta), and DNC (organge) all bound on the protein surface of the binding site of human COMT. Superimposition of NDGA (cyan) and 4-OHE2 (magenta), and NDGA and DNC (orange) at the human COMT active site is shown as a stick representation (C) and (D), respectively. The amino acid residues involved in the key hydrogen bonding interactions are depicted by atom type: carbon atoms in gray, nitrogen atoms in blue, sulfur atoms in yellow, and oxygen atoms in red. Polar hydrogen bonds are shown as yellow dashed lines. A comparative view of the hydrogen bond interactions between COMT and NDGA (cyan) (E), 4-OHE2 (magenta) (F), and DNC (orange) (G) is shown as a stick representation. Numbers presented adjacent to yellow dashed lines represent the calculated length of the hydrogen bond in Å.
